

# Bioinformatics analysis and identification of circular RNAs promoting the osteogenic differentiation of human bone marrow mesenchymal stem cells on titanium treated by surface mechanical attrition

Shanshan Zhu[1,*], Yuhe Zhu[1,*], Zhenbo Wang[2], Chen Liang[2], Nanjue Cao[3], Ming Yan[1], Fei Gao[1], Jie Liu[4] and Wei Wang[1]

[1] School and Hospital of Stomatology, China Medical University, Liaoning Provincial Key Laboratory of Oral Diseases, Shenyang, Liaoning, China
[2] Shenyang National Laboratory for Materials Science, Institute of Metal Research, Chinese Academy of Sciences, Shenyang, Liaoning, China
[3] The Fourth Affiliated Hospital, Zhejiang University School of Medicine, Yiwu, Zhejiang, China
[4] Department 1 of Science Experiment Center, China Medical University, Shenyang, Liaoning, China
* These authors contributed equally to this work.

Corresponding authors
Wei Wang, wwang75@cmu.edu.cn
Jie Liu, lj6152003@163.com

## ABSTRACT

**Background:** To analyze and identify the circular RNAs (circRNAs) involved in promoting the osteogenic differentiation of human bone mesenchymal stem cells (hBMSCs) on titanium by surface mechanical attrition treatment (SMAT).
**Methods:** The experimental material was SMAT titanium and the control material was annealed titanium. Cell Counting Kits-8 (CCK-8) was used to detect the proliferation of hBMSCs, and alkaline phosphatase (ALP) activity and alizarin red staining were used to detect the osteogenic differentiation of hBMSCs on the sample surfaces. The bioinformatics prediction software miwalk3.0 was used to construct competing endogenous RNA (ceRNA) networks by predicting circRNAs with osteogenesis-related messenger RNAs (mRNAs) and microRNAs (miRNAs). The circRNAs located at the key positions in the networks were selected and analyzed by quantitative real-time PCR (QRT-PCR).
**Results:** Compared with annealed titanium, SMAT titanium could promote the proliferation and osteogenic differentiation of hBMSCs. The total number of predicted circRNAs was 51. Among these, 30 circRNAs and 8 miRNAs constituted 6 ceRNA networks. Circ-LTBP2 was selected for verification. QRT-PCR results showed that the expression levels of hsa_circ_0032599, hsa_circ_0032600 and hsa_circ_0032601 were upregulated in the experimental group compared with those in the control group; the differential expression of hsa_circ_0032600 was the most obvious and statistically significant, with a fold change (FC) = 4.25 ± 1.60, $p$-values ($p$) < 0.05.

## INTRODUCTION

Oral implant technology can effectively repair dentition defects. Compared with traditional repair methods, oral implants have more beautiful shape, have stronger retention, result in higher chewing efficiency and lower foreign body sensation, and can delay the absorption of alveolar bone (*Camargo et al., 2017*). At present, dental implants are mainly made of pure titanium and its alloys because titanium has a low density, low modulus, high strength, and excellent biocompatibility and corrosion resistance (*Song et al., 2019*; *Duan et al., 2019*; *Zhou et al., 2019*). However, the corrosion resistance of titanium after implantation in the human body is reduced, resulting in "stress shielding" that causes fibrous binding rather than osseointegration and affecting the stability of the implants (*Sven et al., 2018*; *Ito et al., 2018*; *Minko, Belyavin & Sheleg, 2017*). Previous works showed that surface mechanical attrition treatment (SMAT) technology might be applied to solve this problem by improving osseointegration ability on Ti surface (*Lai et al., 2012*). In addition, SMAT metals have the advantages of high strength (*Fang et al., 2011*), decreased surface alloying temperature/time due to high diffusion rate (*Wang & Lu, 2017*) and high chemical reactivity (*Tong et al., 2003*), and their wear resistance and fatigue resistance are also significantly improved (*Wang et al., 2003*; *Lei et al., 2019*).

Human bone marrow mesenchymal stem cells (HBMSCs) are easy to acquire, isolate, culture and purify. They have strong proliferation and differentiation potential. After many passages, they still have the characteristics of stem cells; they function in hematopoietic support and can secrete a variety of growth factors. They also have an immunoregulatory function and show low immunological rejection characteristics (*Son et al., 2018*; *Miranda et al., 2012*).

Circular RNA is double-stranded non-coding RNA that is bound by a covalent bond into a closed loop structure without a 5′ cap and 3′ poly tail and is not easily degraded by exonucleases. It is more stable than linear mRNA, and its length is between 200 bp and 2,000 bp (*Zhao et al., 2019*; *Ji et al., 2018*); it is highly conserved, and it is found in eukaryotes, mostly within the cytoplasm. A total of 16% of circRNAs are derived from coding genes, while 85% are derived from exon rings. CircRNAs have specific reverse cleavage sites, and multiple circRNAs can be produced at the same locus; this mainly occurs at the post-transcriptional level, and most transcripts are non-encoding (*Liu et al., 2017b*; *Zhao et al., 2018*; *Zhang & Xin, 2018*). CircRNAs show specificity in terms of timing, organization, and disease specificity. They accumulate in the nerve tissue during aging and can be secreted from the outside of the cell to form a ring in the membrane. The main types of circRNAs include exon circRNAs, exon-intron circRNAs, intron circRNAs, antisense circRNAs, intergenic circRNAs, and sensory-overlap circRNAs. The main functions of circRNAs are as follows: to function as ceRNA molecules (as miRNA molecular "sponges") to bind to miRNA response elements (MREs) to reduce the effects of miRNA on gene expression; to regulate classical RNA splicing so that classical RNA is inhibited; to regulate parental gene transcription (*Kulcheski, Christoff & Margis, 2016*; *Cortés-López et al., 2018*); to bind to RNA-binding proteins to regulate other

RNAs; to be translated into proteins; to induce pseudogene expression after reverse transcription (*Li & Han, 2019*).

A large number of studies in recent years have shown that circRNA can regulate stem cell osteogenic differentiation. *Mengjun, Lingfei & Yunfei (2018)* performed microarray analysis to determine the expression profile of circRNAs during osteoblast differentiation. The results indicated that the functional annotation of differentially expressed circRNAs was associated with osteogenic differentiation. The researchers then constructed a circRNA-miRNA network, and network analysis indicated that some circRNAs were associated with miRNAs with osteogenic effects. In addition, the researchers verified the expression of a central miRNA, miR-199b-5p, and its associated circRNA circIGSF11. The results showed that silencing circIGSF11 can promote osteoblast differentiation; *Zheng et al. (2017)* induced periodontal ligament stem cells (PDLSCs), which were analyzed by circRNA sequencing, qRT-PCR, differential expression analysis and gene ontology (GO) analysis. The target mRNAs regulated by the differentially expressed circRNAs were enriched in cell matrix formation and osteogenic differentiation.

This study intends to use biological information prediction to analyze the interactions between miRNAs, circRNAs, and circRNA-miRNA-mRNA that interact with osteogenic mRNAs. It also seeks to investigate whether the SMAT titanium material can promote hBMSC osteogenic differentiation by affecting circRNA and provide a theoretical basis for the application of SMAT in oral clinic.

## MATERIALS AND METHODS

### Preparation of titanium sheets

Experimental group: Pure titanium plates in medical grade, in a cylindrical shape with a diameter of 60 mm and a thickness of 5 mm, were studied in this work. We performed ultrasonic-assisted SMAT on the plate samples by using a SNC-2 machine (New Nano-crystal Technology Co., Ltd., China) at a frequency of 20,000 Hz. During SMAT, hardened steel balls with a diameter of 6 mm were driven by the system and repeatedly impacted onto the sample surface for 30 min, so that a gradient nanostructured surface layer was formed. The SMAT plates were electrically spark-cut into discs with a diameter of 11 mm and a thicknesses of 2.5 mm, as the samples in the experimental group.

Controlled group: An annealing treatment was carried out at 680 °C for 2 h on the SMAT samples to prepare the samples in the controlled group. By doing this, the surface morphologies of the experimental and controlled samples were similar, while the grains in the surface layer of the controlled samples became coarse.

Samples in both the experimental and controlled groups were sequentially cleaned in acetone, 36–38% dilute hydrochloric acid, absolute ethanol and distilled water, and then ultrasonically washed for 20 min, dried and autoclaved prior to use.

Transmission electron microscope (TEM; JEOL, Japan) was used to observe the surface structure of the experimental materials so as to determine whether it has gradient nanostructures.

## Culture of hBMSCs

HBMSCs were purchased from Beijing Yuhengfeng Technology Co., Ltd. We resuscitated the 10th generation of hBMSCs and placed them in α-modified Eagle medium (αMEM, HyClone) containing 15% foetal bovine serum (FBS, HyClone) and 100 U/ml double antibody (HyClone), and the cells were cultured at 37 °C in a 5% $CO_2$ cell culture incubator. The cells were in the logarithmic phase of growth during all experiments.

## CCK-8 method for determining the cell proliferation curve

The 14th generation hBMSCs showing good growth were inoculated at $4 \times 10^4/cm^2$ on SMAT titanium and annealed titanium, which were placed in 24-well plates. Each group of materials was placed in 3 wells. Each well was filled with 1 ml of basic medium. A total of 100 µl of CCK-8 (KGI Biotechnology Co., Ltd., China) was added to each well on day 1, 3, 5, and 7. We incubated the plates for 2 h at 37 °C. Then, we pipetted 110 µl of reaction solution per well into a 96-well plate. A microplate reader (Tecan, Männedorf, Switzerland) was used to detect the optical density (OD) value of each well at a wavelength of 450 nm. The proliferation curve of the 14th generation of hBMSCs was generated with time as the horizontal axis and the average OD as the vertical axis.

## ALP activity detection

We inoculated 14th generation hBMSCs showing good growth at $4 \times 10^4/cm^2$ on SMAT titanium and annealed titanium, which were placed in 24-well plates. Each group of materials was placed in 3 wells, and each well was supplemented with 1 ml basic culture solution. When the cell fusion degree was 80–90%, we added osteogenic induction medium (15% FBS, 1% streptomycin mixture, $10^{-7}$ M dexamethasone, $10^{-2}$ β-glycerophosphate disodium salt, 50 µg/ml Vitamin C, αMEM) and cultured the cells in a 37 °C incubator. After 3, 5, 7, and 14 days of culture, we digested the cells of each group with trypsin (Solarbio). We lysed the cells with 1% Triton X-100 (Solarbio) and repeatedly thawed them until they were disrupted, and then we collected the supernatant. The ALP activity of each group of cells was determined and calculated according to the instructions in the ALP activity test kit (Institute of Bioengineering, Nanjing, China) and Bradfords (BCA) kit (Institute of Bioengineering, Nanjing, China).

## Alizarin red staining

The 14th generation hBMSCs showing good growth were inoculated at $4 \times 10^4/cm^2$ on SMAT titanium and annealed titanium, which were placed in 24-well plates. Each group of materials was plated in 3 wells supplemented with 1 ml basic culture solution. When the cell fusion degree was 80–90%, we added osteogenic induction medium and cultured the cells in a constant temperature incubator. After 7, 14, and 21 days of culture, we fixed the cells in 95% ethanol for 1 h and stained them with 1% Alizarin red (Sigma) for 2 h.

## Biological information prediction

### Prediction and screening of osteogenesis-related miRNAs

The osteogenesis-related literature from 2015 to 2019 was used to select 20 osteogenesis-associated mRNAs. Twelve mRNAs associated with the transforming growth factor-β (TGF-β)/drosophila mothers against decapentaplegic protein (Smad), mitogen-activated protein kinase (MAPK)/extracellular signal-regulated kinase (ERK), Wnt, and Notch signaling pathways were analyzed and screened in combination with Kyoto Encyclopaedia of Genes and Genomes (KEGG) signaling pathway analysis. MiRNAs 3.0 software was used to screen for miRNAs capable of targeting these candidate mRNAs. The underlying screening principle involved the identification of miRNAs that were located in the seed area as predicted by TargetScan, miRDB and miwalk3.0.

### The prediction and screening of osteogenesis-related circRNAs

The formation of circRNA is based on exon cyclization. Miwalk3.0 software was used to screen the complementary sequence of the coding sequence (CDS) region of the obtained miRNA, and the mRNA sequence of the seed region was used as the source transcript for the circRNA. The exon of each gene was searched based on the complementary region, and the circRNA including the exon was screened. The length of the circRNAs selected for the experiment were 200–2,000 bp, and each was detected in more than 2 samples from the circBase.

### Predictive analysis of ceRNA networks of osteogenesis-related circRNAs

The mRNA-miRNA-circRNA interactions were analyzed to determine the frequency distribution, and network analysis was performed by using Cytoscape software to identify the core molecules and construct the ceRNA networks. The interactions between them were analyzed, and the selected circRNAs located at key locations in the networks were used for qRT-PCR verification.

## QRT-PCR detection of the differential expression of circRNAs during the osteogenic differentiation of hBMSCs on the two groups of titanium plates

The 14th generation hBMSCs showing good growth were inoculated at $4 \times 10^5$/cm$^2$ on SMAT titanium and annealed titanium in 6-well plates. We used 3 wells for each set of materials, and each well was filled with 3 ml of basic culture solution. When the cell fusion degree was 80–90%, we added osteogenic induction medium and cultured the cells in a constant temperature incubator. Once the cells had been osteogenically induced for 3, 5, 14, 21 days, two sets of total RNA were separately extracted with TRIzol (Gibco, Dublin, Ireland), and the purity and amount of the collected RNA were determined by a Nanodrop 2000 microultraviolet analyzer (Thermo Company, Waltham, MA, USA). CDNA was synthesized using the PrimeScript RT Master Mix (Takara, Tokyo, Japan). Shengong Biological Engineering Co., Ltd. (Shanghai, China) designed specific primers for the circRNAs. QRT-PCR was performed using TB Green Premix EX TaqTM II (Takara, Tokyo, Japan), and PCR-specific amplification was conducted with a Prism 7500 real-time
**Table 1 Primer and internal reference gene sequence table.**

| Name of circRNA/internal reference | Primer sequence (5′–3′) | Product length (bp) |
|---|---|---|
| hsa_circ_0032599 | CTGGAACCTGCGTGAACCT | 226 |
| | TGATCCGCTGGGCCAAAG | |
| hsa_circ_0032600 | CTGGAACCTGCGTGAACCT | 221 |
| | TCTTGGCAGTGAGTGAGGTT | |
| hsa_circ_0032601 | CTGGAACCTGCGTGAACCT | 229 |
| | TGGTGGATCTGCACTGAGG | |
| GAPDH | GGACCTGACCTGCCGTCTAG | 99 |
| | TAGCCCAGGATGCCCTTGAG | |

PCR instrument (ABI Company, Tampa, FL, USA). The expression of circRNAs was determined based on the threshold cycle (Ct), and the relative expression levels were calculated using the $2^{-\Delta\Delta Ct}$ method. GAPDH served as an internal standard control. The primer and internal reference fragment sequences (5′–3′) are as Table 1.

## Statistical analysis

All experiments were repeated three times. SPSS Statistics 20.0 software and GraphPad Prism 7 were used to perform the statistical analysis. We used one-way ANOVA and the LSD-$t$ test to compare data between groups. Data are expressed as the mean ± standard deviation, and the differences with a FC ≥ 2.0 and a $p < 0.05$ were considered statistically significant.

# RESULTS

## Preparation of titanium

The TEM observation shows that the SMAT method can make the surface of gradient nano-metal pure titanium nanocrystallization, and the grain size is on the nanometer scale (Fig.1).

## Culture of hBMSCs

HBMSCs were observed as adherent cells under an inverted phase contrast microscope. One day after passage, the cells were partially attached (Fig. 2A); after 3 days, most of the cells were adherent, and the cell fusion degree was over 80%. The morphology of the adherent cells changed from round to long fusiform or star-shaped, and the arrangement was spiral (Fig. 2B).

## Proliferation curve of the hBMSCs

(1) After observing the hBMSC subculture for 7 days, the results showed that the hBMSCs showed obvious latency, logarithmic proliferative and plateau phases, and the growth curve assumed an "S" shape. Days 1–3 were the incubation period, and days 3–5 were the logarithmic growth phase; On the 3rd and 5th day of culture, the difference between the SMAT titanium group and the annealed titanium group was statistically significant ($p < 0.05$; Fig. 2C).

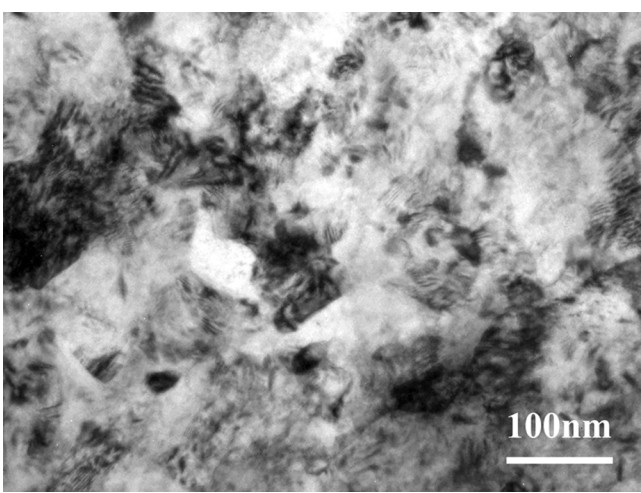

**Figure 1  The TEM observation of the gradient nano-metal pure titanium.**

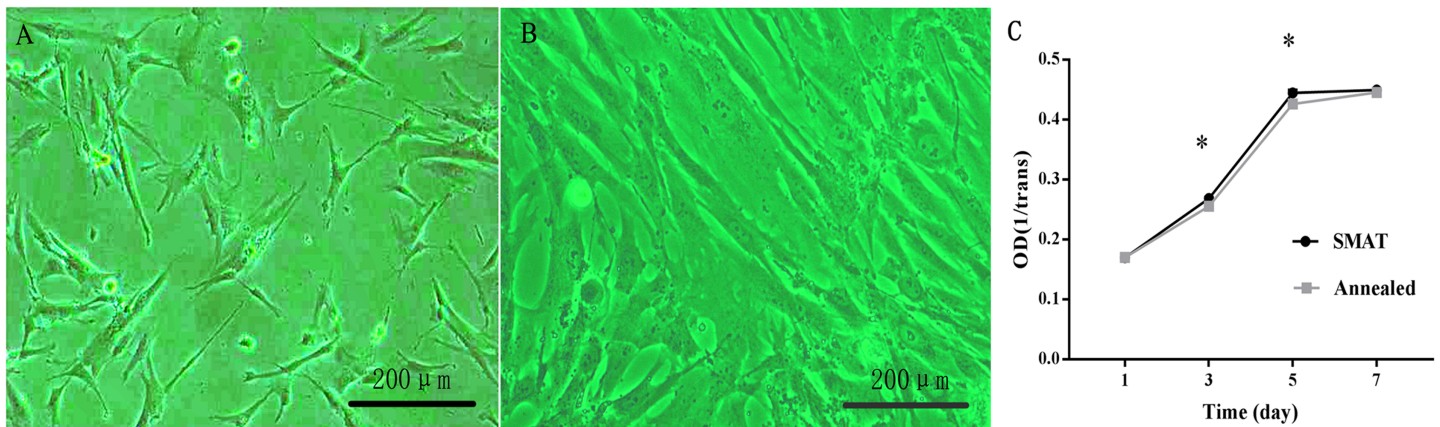

**Figure 2  Culture and proliferation curve of the hBMSCs.** (A) HBMSCs cultured for 1 day (×100). (B) HBMSCs cultured for 3 days (×100). (C) CCK8 detection at 1, 3, 5, and 7 days of hBMSCs culture. *The SMAT group compared with the annealed group (mean ± SD, $n = 3$, *indicates $p < 0.05$).

## Detection of ALP activity

When hBMSCs were cultured for 3, 5, 7 and 14 days, the ALP activity level in each group increased over time. At each time point, the level of ALP activity in the SMAT titanium group was higher than that in the annealed titanium group. On days 3 and 5, the differences between the SMAT titanium group and the annealed titanium group were statistically significant ($p < 0.05$; Fig. 3A). It is shown that SMAT titanium can promote the early osteogenic differentiation of hBMSCs.

## Alizarin red staining

HBMSCs were induced osteogenically for 7, 14 and 21 days, and the alizarin red staining area of each group increased over time. At each time point, the alizarin red staining area of the SMAT titanium group was larger than that of the annealed titanium group. On the

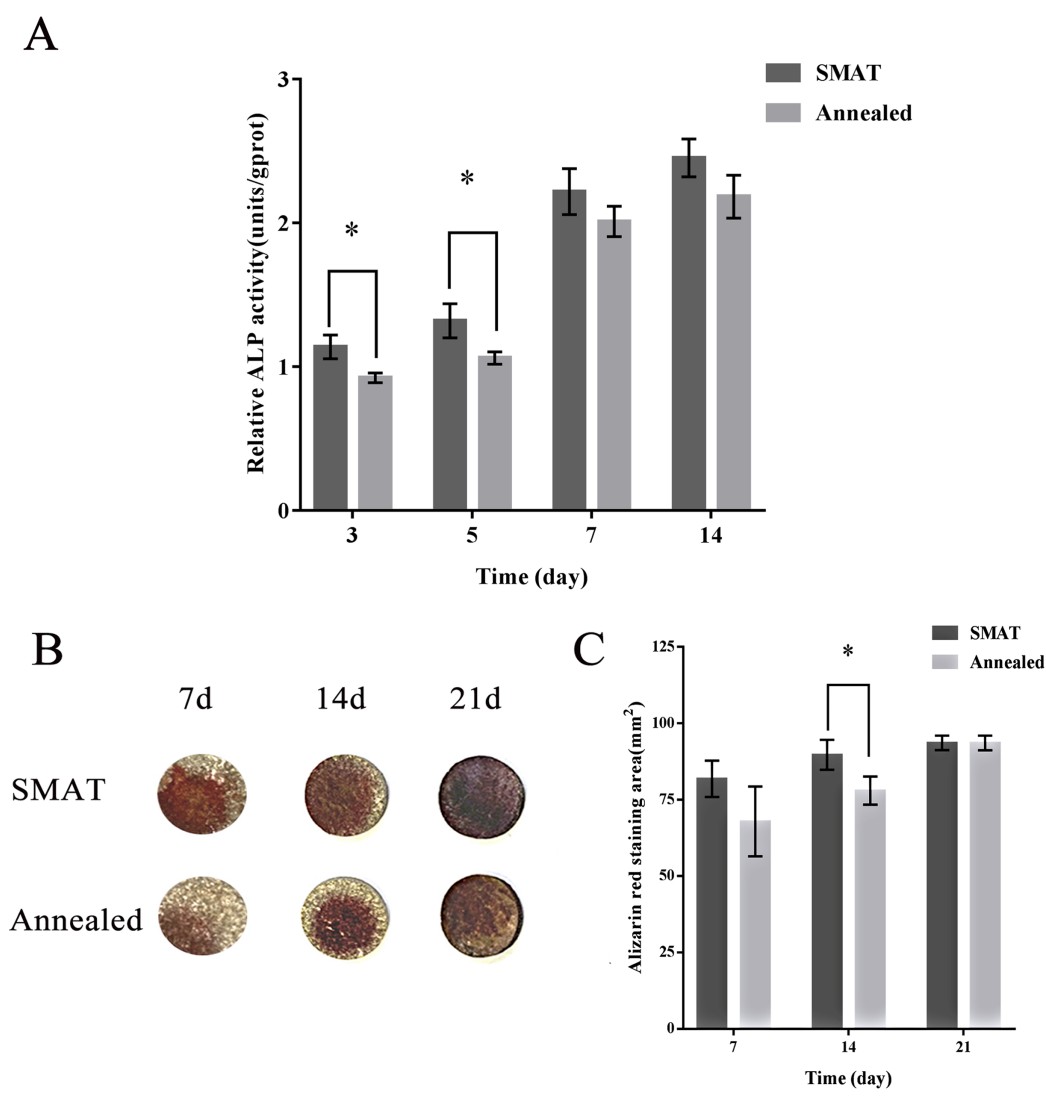

**Figure 3 Detection of ALP activity and alizarin red staining of hBMSCs.** (A) Detection of ALP activity at 3, 5, 7 and 14 days after osteogenic induction of hBMSCs. *The SMAT group compared with the annealed group (mean ± SD, $n = 3$, *indicates $p < 0.05$). (B) Alizarin red staining after osteogenic induction of hBMSCs for 7, 14 and 21 days. (C) The area of alizarin red staining at 7, 14, 21 days after osteogenic induction of hBMSCs. *The SMAT group compared with the annealed group (mean ± SD, $n = 3$, *indicates $p < 0.05$).

14th day, the difference between the SMAT titanium group and the annealed titanium group was statistically significant ($p < 0.05$; Figs. 3B and 3C). It is shown that SMAT titanium can promote the late osteogenic differentiation of hBMSCs.

## Biological information prediction
### *Screening results for the osteogenesis-related miRNAs*
According to the literature, 20 osteogenesis-related mRNAs have been identified, including ALK, COLIA1, RUNX2, SP7, IBSP, COL10VA1, BMP2, BGLAP, SPP1, DLX5, CBFB, BMP7, AcvR1b, LIF, INHBA, Noda1, BMP4, FGF2, IGF1, Wnt. Twelve mRNAs were

confirmed to be involved in the TGF-β/Smad, MAPK/ERK, Wnt, and Notch signaling pathways, including Wnt3, Wnt4, Wnt8B, RUNX2, IGF1, ACVR1B, INHBC, CBFB, LIF, BMP7, NOG, CHRD. Among them, Wnt3, Wnt4 and Wnt8B genes are involved in the Wnt signaling pathway, the RUNX2 gene is involved in the Notch signaling pathway, the CBFB gene is involved in the Smad2/3 signaling pathway, the LIF gene is involved in theERK signaling pathway, the IGF1 is involved in the MAPK/ERK signaling pathway, the INHBC, ACVR1B, NOG, CHRD genes are involved in the TGF-β signaling pathway, and the BMP7 gene is involved in the TGF-β and ERK signaling pathways (Figs. 4C–4L).

Eight miRNAs were predicted, including hsa-miR-18b-5p, hsa-let-7b-5p, hsa-miR-1224-5p, hsa-miR-129-5p, hsa-miR-145-5p, hsa-miR-24-3p, hsa-miR-5195-3p, hsa-miR-6088 (Table 2).

### Screening results for theosteogenesis-related circRNAs

Fifty-one circRNAs were screened by miwalk3.0 software (Table 3; Figs. 4A and 4B).

### CeRNA network analysis of the osteogenesis-related circRNAs

Further analysis of the predicted 51 circRNAs revealed that 30 circRNAs and 8 miRNAs constituted 6 ceRNA networks. Circ-GNB5/circ-HERC1/circ-KMT2A/circ-LTBP2 interact with hsa-miR-24-3p and Wnt3 gene in the Wnt signaling pathway; circ-TBC1D2B/circ-TBCD/circ-TRIOBP/circ-VPS13C interact with hsa-miR-6088 and CBFB gene in the Smad2/3 signaling pathway; circ-ABCA3 interacts with hsa-miR-18b-5p and CF2L2; circ-MDN1/circ-MYH9/circ-PLPPR4/circ-RAB31/circ-SMARCH2/circ-SPEG/circ-SYNM/circ-TANGO6 interact with hsa-miR-5195-3p and Wnt4 gene in the Wnt signaling pathway, the IGF1 gene in the MAPK/ERK signaling pathway; hsa-miR-1224-5p combines with NOG gene in the TGF-β signaling pathway; circ-ADAMTS13/circ-ARHGAP32/circ-BRPF3/circ-CARD8/circ-CCDC88C/circ-COL4A2/circ-DGKD/circ-DIP2C/circ-DYNC1H1/circ-FAT3/circ-LTBP2 interact with hsa-let-7b-5p and CHRD gene in the TGF-β signaling pathway (Fig. 4M).

## QRT-PCR assay

To verify the accuracy of the bioinformatics prediction, we screened circ-LTBP2 expression by using QRT-PCR detection. The results of QRT-PCR showed that the expression levels of hsa_circ_0032599, hsa_circ_0032600 and hsa_circ_0032601 were upregulated in the experimental group, and the differential expression of hsa_circ_0032600 was the most upregulated and statistically significant on the 14th day (FC= $4.25 \pm 1.60$, $p < 0.05$; Fig. 5).

## DISCUSSION

SMAT technology uses high-energy balls to repeatedly impact the sample surface. Typically, a gradient nanostructured surface layer, in which the grain size is in the nanometer scale at the top surface and gradually increases into the micrometer scale in the interior, will be achieved on metals by SMAT (Fang et al., 2011; Lu & Lu, 2004). In this study, the TEM observation shows that the SMAT method can make the surface of

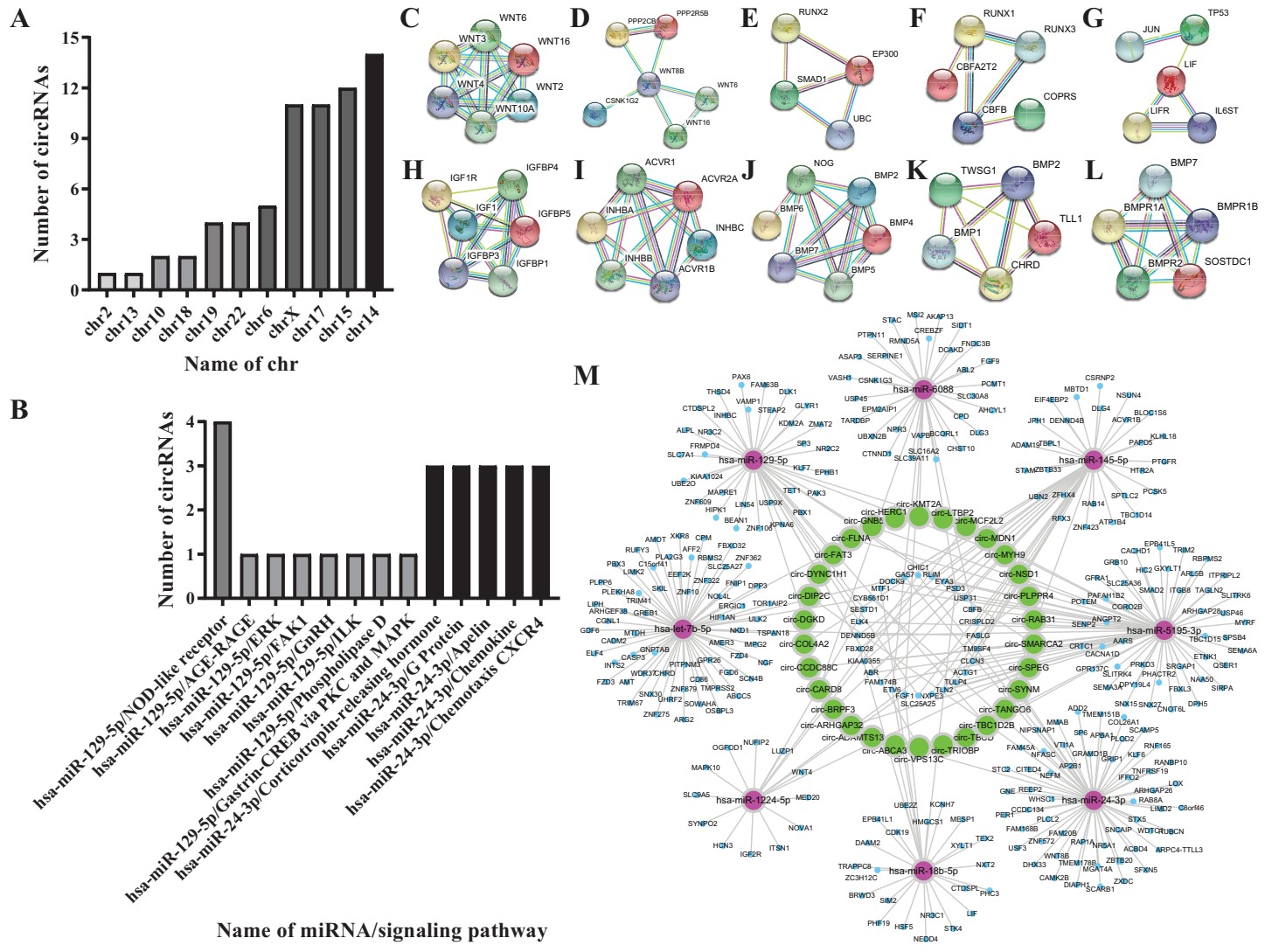

**Figure 4 Biological information prediction.** (A) Numbers of circRNAs corresponding to different chrs. (B) Numbers of circRNAs on different signaling pathways. (C) WNT3, WNT4 genes in Wnt signaling pathway. (D) WNT8B gene in Wnt signaling pathway. (E) RUNX2 gene in Notch signaling pathway. (F) CBFB gene in Smad2/3 signaling pathway. (G) LIF gene in ERK signaling pathway. (H) IGF1 gene in MAPK/ERK signaling pathway. (I) INHBC, ACVR1B genes in TGF-βsignaling pathway. (J) NOG gene in TGF-βsignaling pathway. (K) CHRD gene in TGF-β signaling pathway. (L) BMP7 gene in TGF-β, ERK signaling pathways. (M) Network analysis of osteogenesis-related miRNAs and their interacting circRNAs and target genes.

gradient nano-metal pure titanium nanocrystallization, and the grain size is on the nanometer scale. Since there is no interface separating the surface layer and the substrate, the problem of poor interface bonding between the nanostructured surface layer and the substrate, usually caused by traditional coating techniques, is easily solved (*Nana & Ning, 2018*; *Du et al., 2019*). In addition, SMAT materials have the advantages of high strength, high hardness, high diffusion rate and high chemical reactivity, and their wear resistance and fatigue resistance are also significantly improved (*Nowak, Serafin & Wierzba, 2019*; *Benafia, Retraint & Brou, 2018*; *Yao et al., 2017*; *Fang et al., 2011*; *Wang & Lu, 2017*; *Tong et al., 2003*; *Wang et al., 2003*; *Lei et al., 2019*; *Lu & Lu, 2004*). In this study,

**Table 2 Predicted miRNAs and their gene symbols and signaling pathways.**

| miRNA | Gene symbol | Signaling pathway |
|---|---|---|
| hsa-miR-18b-5p | ABCA3 | |
| hsa-let-7b-5p | HERC1 | |
| hsa-miR-1224-5p | MCF2L2, PLPPR4, SPEG | |
| hsa-miR-129-5p | FAT3, LTBP2, MYH9, SMARCA2, TBC1D2B, VPS13C, DYNC1H1 | |
| | DGKD | Phospholipase D, Gastrin-CREB via PKC and MAPK |
| hsa-miR-145-5p | BRPF3, TANGO6 | |
| | COL4A2 | AGE-RAGE, ERK, FAK1, GnRH, ILK |
| hsa-miR-24-3p | ADAMTS13, DIP2C, SYNM | |
| hsa-miR-5195-3p | ARHGAP32, CCDC88C, FLNA, LTBP2, MDN1, RAB31, TBCD, TRIOBP | |
| | GNB5 | Corticotropin-releasing hormone, G Protein, Apelin, Chemokine, Chemotaxis CXCR4 |
| hsa-miR-6088 | CARD8 | NOD-like receptor |
| | KMT2A | |
| hsa-miR-18b-5p | ABCA3 | |
| hsa-let-7b-5p | HERC1 | |
| hsa-miR-1224-5p | MCF2L2, PLPPR4, SPEG | |
| hsa-miR-129-5p | FAT3, LTBP2, MYH9, SMARCA2, TBC1D2B, VPS13C, DYNC1H1 | |
| | DGKD | Phospholipase D, Gastrin-CREB via PKC and MAPK |
| hsa-miR-145-5p | BRPF3, TANGO6 | |
| | COL4A2 | AGE-RAGE, ERK, FAK1, GnRH, ILK |
| hsa-miR-24-3p | ADAMTS13, DIP2C, SYNM | |
| hsa-miR-5195-3p | ARHGAP32, CCDC88C, FLNA, LTBP2, MDN1, RAB31, TBCD, TRIOBP | |
| | GNB5 | Corticotropin-releasing hormone, G Protein, Apelin, Chemokine, Chemotaxis CXCR4 |
| hsa-miR-6088 | CARD8 | NOD-like receptor |
| | KMT2A | |

the influence of surface morphologies on the adhesion, proliferation and differentiation of hBMSCs might be safely excluded, while the samples in the experimental and controlled groups show a similar surface morphology (with surface roughness $R_a$ of 1.73 ± 0.11 and 1.82 ± 0.06 μm, respectively). Therefore, it should be the contribution from the surface nanostructure that the hBMSC osteogenic differentiation is promoted on the SMAT titanium.

In this study, hBMSCs can differentiate into osteoblasts. A lot of literature have confirmed that hBMSCs are important members of a family of stem cells and are derived from early developing mesoderm and ectoderm. They are easy to extract and purify, and

**Table 3 Predicted circRNAs and their chrs, gene symbols, miRNAs, and signaling pathways.**

| chr | hsa_circRNA | Gene symbol | miRMA | Signaling pathway |
|---|---|---|---|---|
| chr10 | hsa_circ_0076184 | BRPF3 | hsa-miR-129-5p | |
| | hsa_circ_0002749 | CARD8 | hsa-miR-129-5p | NOD-like receptor |
| chr13 | hsa_circ_0051734 | CARD8 | hsa-miR-129-5p | NOD-like receptor |
| chr | hsa_circRNA | Gene symbol | miRMA | Signaling pathway |
| chr14 | hsa_circ_005173 | CARD8 | hsa-miR-129-5p | NOD-like receptor |
| | hsa_circ_0051736 | | | |
| | hsa_circ_0030878 | COL4A2 | hsa-miR-129-5p | AGE-RAGE, ERK, FAK1, GnRH, ILK |
| | hsa_circ_0058776 | DGKD | hsa-miR-129-5p | Phospholipase D, Gastrin-CREB viaPKC and MAPK |
| | hsa_circ_0017364 | DIP2C | hsa-miR-129-5p | |
| | hsa_circ_0017365 | | | |
| | hsa_circ_0091990 | | | |
| | hsa_circ_0091991 | | | |
| | hsa_circ_0091992 | FLNA | hsa-miR-145-5p | |
| | hsa_circ_0091997 | | | |
| | hsa_circ_0091998 | | | |
| | hsa_circ_0091999 | | | |
| | hsa_circ_0092000 | | | |
| | hsa_circ_0092001 | | | |
| chr15 | hsa_circ_0092008 | FLNA | hsa-miR-145-5p | |
| | hsa_circ_0092009 | | | |
| | hsa_circ_0092010 | | | |
| | hsa_circ_0035311 | GNB5 | hsa-miR-24-3p | Corticotropin-releasing hormone, G Protein, Apelin, Chemokine, Chemotaxis CXCR4 |
| | hsa_circ_0035312 | | | |
| | hsa_circ_0035313 | | | |
| | hsa_circ_0032566 | LTBP2 | hsa-miR-24-3p | |
| | hsa_circ_0032576 | | | |
| | hsa_circ_0032579 | | | |
| | hsa_circ_0032584 | | | |
| | hsa_circ_0032585 | | | |
| | hsa_circ_0032594 | | | |
| chr17 | hsa_circ_0032599 | | | |
| | hsa_circ_0032600 | LTBP2 | hsa-miR-24-3p | |
| | hsa_circ_0032601 | | | |
| | hsa_circ_0032602 | | | |
| | hsa_circ_0032604 | | | |
| | hsa_circ_0032605 | | | |
| | hsa_circ_0032607 | | | |
| | hsa_circ_0032608 | | | |

| chr | hsa_circRNA | Gene symbol | miRMA | Signaling pathway |
|---|---|---|---|---|
| | | **Table 3** (continued) | | |
| chr17 | hsa_circ_0077356 | MDN1 | hsa-miR-5195-3p | |
| | hsa_circ_0077360 | | | |
| | hsa_circ_0077371 | | | |
| chr18 | hsa_circ_0077375 | MDN1 | hsa-miR-5195-3p | |
| | hsa_circ_0063112 | MYH9 | hsa-miR-5195-3p | |
| chr19 | hsa_circ_0063114 | MYH9 | hsa-miR-5195-3p | |
| | hsa_circ_0063115 | | | |
| | hsa_circ_0063117 | | | |
| | hsa_circ_0046890 | RAB31 | | |
| chr2 | hsa_circ_0046891 | RAB31 | | |
| chr22 | hsa_circ_0037043 | SYNM | hsa-miR-5195-3p | |
| | hsa_circ_0036435 | TBC1D2B | hsa-miR-6088 | |
| | hsa_circ_0036436 | | | |
| | hsa_circ_0036437 | | | |
| chr6 | hsa_circ_0036438 | TBC1D2B | hsa-miR-6088 | |
| | hsa_circ_0036439 | | | |
| | hsa_circ_0036440 | | | |
| | hsa_circ_0036441 | | | |
| | hsa_circ_0046529 | TBCD | hsa-miR-6088 | |
| Chrx | hsa_circ_0046568 | TBCD | hsa-miR-6088 | |
| | hsa_circ_0046569 | | | |
| | hsa_circ_0046577 | | | |
| | hsa_circ_0046588 | | | |
| | hsa_circ_0046589 | | | |
| | hsa_circ_0046590 | | | |
| | hsa_circ_0046591 | | | |
| | hsa_circ_0046594 | | | |
| | hsa_circ_0046595 | | | |
| | hsa_circ_0046596 | | | |
| | hsa_circ_0035596 | VPS13C | hsa-miR-6088 | |

they still show stem cell characteristics after repeated subculturing. They have strong proliferation and differentiation ability and can be multi-directionally differentiated into osteoblasts, chondrocytes, hematopoietic cells, muscle cells and other types of cells. They can be used to repair tissue and organ damage (*Saito et al., 2018*; *Mohammadali, Abroun & Atashi, 2018*). They have a hematopoietic support function, since they not only provide mechanical support for hematopoietic stem cells in the bone marrow but also secrete a variety of growth factors, such as interleukin (IL)-6, IL-11, leukemia inhibitory factor (LIF), macrophage colony-stimulating factor (M-CSF) and human stem cell growth factor (SCF), to support hematopoiesis (*Amini et al., 2018*; *Mohammadali, Abroun & Atashi, 2018*). HBMSCs also regulate immune function and do

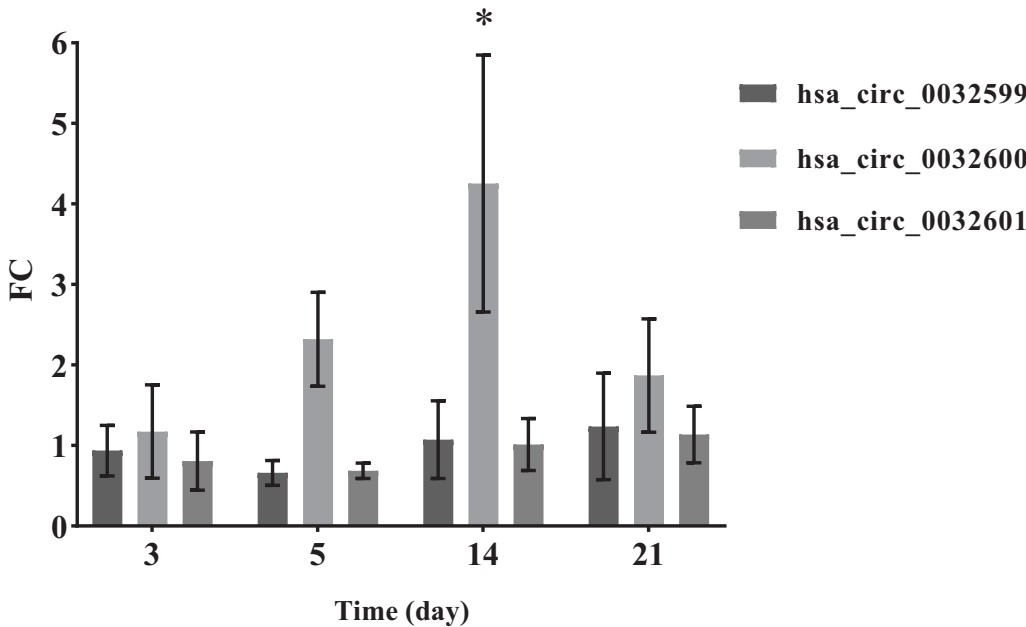

**Figure 5 QRT-PCR results of the three differentially expressed circRNAs.** *The SMAT group compared with the annealed group, FC > 2.0 (mean ± SD, $n$ = 3, *indicates $p < 0.05$).

not show immunological characteristics consistent with rejection (*Liu et al., 2018*; *Gabr et al., 2017*; *Lee et al., 2017*).

To date, several osteogenesis-related signaling pathways have been documented, including the TGF-β, Smad, MAPK, Wnt/β-catenin, Notch, Hedgehog, fibroblast growth factor (FGF), and orthopantomography (OPG)/receptor activator for nuclear factor-κB ligand (RANKL) signaling pathways (*Zhu et al., 2018*; *Urbanek et al., 2017*; *Jin et al., 2017*). TGF-β can increase intracellular ALP activity and the synthesis and secretion of osteocalcin, collagen, and osteonectin (*Yang et al., 2018*). The TGF-β pathway consists of extracellular ligands, transmembrane receptors, and intracellular regulatory factors. The receptor TGF-β activates Smads and enters the nucleus, thereby regulating its target genes (*Hui et al., 2018*); the MAPK signaling pathway involves MAPK/ERK/big MAP kinase 1 (BMK1)/stress activated protein (SAPK)/c-Jun N-terminal kinase (JNK)/P38 in five main ways (*Lou et al., 2019*; *Jiang et al., 2018*). The MAPK signaling pathway also plays an important role in the proliferation, differentiation and apoptosis of osteoblasts (*Li et al., 2018*). Wnt signaling plays an important role in the proliferation and differentiation of mesenchymal stem cells, osteoblastogenesis, bone formation, bone remodeling, and other processes (*Wang et al., 2019*; *Shuai et al., 2019*). The Wnt signaling pathway can also be involved in cell-to-cell signaling through paracrine or autocrine signaling and participates in a variety of cellular processes, such as cell proliferation, differentiation, polarization, and migration (*Hang et al., 2019*; *Liu et al., 2017a*). The Notch signaling pathway involves the Notch receptor, DSL protein (ligand), DNA binding protein and Notch regulatory molecule. It can play a role in multiple cell morphogenetic processes, such as cell formation, differentiation, and apoptosis (*Nandagopal et al., 2018*;

Sun et al., 2018; Carra et al., 2017). In this study, we choose twelve osteogenesis-relatedmRNAs associated with theTGF-β/Smad, MAPK/ERK, Wnt, and Notch signaling pathways to predict osteogenesis-relatedmiRNAs and circRNAs.

This study proves that there are differences in the expressions of osteogenesis-relatedcircRNAs in the process of SMAT titanium promoting the osteogenic differentiation of hBMSCs. Currently, a large number of studies have shown that circRNA can regulate stem cell osteogenic differentiation. Gu et al. (2017) cultured PDLSCs in osteogenic induction medium and normal medium and found differences in the expression of a total of 1456 circRNAs by RNA sequencing. The researchers used GO and KEGG analyses to predict that circRNA-BANP and circRNA-ITCH can regulate the osteogenic differentiation of PDLSCs through the MAPK pathway. Xiaobei et al. (2018) used qRT-PCR to detect the expression level of CDR1as during the osteogenic differentiation of PDLSCs and constructed CDR1as overexpression and silencing models to detect osteogenic differentiation. Animal experiments were performed to analyze the formation of new bone by microcomputed tomography and various staining methods. The results showed that CDR1as was upregulated and that silencing CDR1as could inhibit the osteogenic differentiation of cells. In vivo experiments have shown that silencing CDR1as can inhibit bone formation.

Alkaline phosphatase is a sugar protease released by cytoplasmic special particles. It hydrolyzes phosphate esters during osteogenesis, provides phosphoric acid for hydroxyapatite deposition, and hydrolyzes pyrophosphate to relieve its inhibitory effect on bone salt formation. Its activity level is positively correlated with the degree of osteogenic differentiation of cells. It is an early marker of osteogenic differentiation of hBMSCs (Samuel et al., 2016; Li et al., 2019). The sample was reacted in a carbonate buffer (pH = 9.8) containing p-nitrobenzene phosphate. Under the action of alkaline phosphatase in the sample, p-nitrobenzene phosphate was decomposed into p-nitrophenol and phosphoric acid. The resulting p-nitrophenol is basic and yellow. Based on measuring the absorbance at 405 nm, the alkaline phosphatase activity in the sample was calculated. The hBMSCs on each group of materials gradually increased ALP activity after osteogenesis induction, and the ALP activity of the SMAT group was higher than that of the annealed group at each time point. On the 3rd and 5th days, the difference between the SMAT group and the annealed group was statistically significant ($p < 0.05$). It is shown that nanostructure can promote the early osteogenic differentiation of hBMSCs. Alizarin red is an anionic dye that forms a red complex with metal ions. It can recognize and chelate calcium ions to form orange-red calcium nodules. It is a sign of late osteogenic differentiation (Li et al., 2019). The area of alizarin red staining of cells on both groups of materials gradually increased with the increase of culture time. At each time point of detection, the alizarin red staining area of the SMAT group was larger than that of the annealed group. This shows that SMAT titanium has better ability to promote the osteogenic differentiation of hBMSCs in late stage than annealed titanium. On the 14th day, the difference between the SMAT group and the annealed group was statistically significant ($p < 0.05$). It is shown that nanostructures can promote the late osteogenic differentiation of hBMSCs.

In the predicted ceRNA networks, one combination of circ-GNB5, circ-HERC1, circ-KMT2A, circ-LTBP2, hsa-miR-24-3p and the Wnt3 gene, another comprised of circ-LTBP2, circ-MCF2L2, circ-MDN1, circ-MYH9, circ-PLPPR4, circ-RAB31, circ-SMARCH2, circ-SPEG, circ-SYNM, circ-TANGO6, hsa-miR-5195-3p and the Wnt4, IGF1 genes, the third consists of circ-ADAMTS13, circ-ARHGAP32, circ-BRPF3, circ-CARD8, circ-CCDC88C, circ-COL4A2, circ-DGKD, circ-DIP2C, circ-DYNC1H1, circ-FAT3, circ-LTBP2, hsa-let-7b-5p and CHRD, three networks communicated with each other through the core hub circ-LTBP2. Based on the previous prediction, it's target mRNAs including wnt3, wnt4, CBFB, IGF1 and CHRD are recognized osteogenic marker genes (*Kureel et al., 2017*; *Qin et al., 2015*; *Lindsey, Rundle & Mohan, 2018*). What's more, circ-LTBP2 has not been reported yet. So we selected circ-LTBP2 for verification.

In the comparison of the two groups of materials, QRT-PCR results showed that the differential expression of hsa_circ_0032600 on the 14th day was significant (*p*-values), which is consistent with the results of alizarin red staining. The combination mRNAs of hsa_circ_0032600 including wnt3, wnt4, CBFB, IGF1, CHRD involved in Wnt, Smad2/3, MAPK/ERK, TGF-β osteogenic pathways, which can regulate osteogenic differentiation in the middle and late stages (*Mobini et al., 2017*; *Li et al., 2019*; *Mao, Bian & Shen, 2012*). So hsa_circ_0032600 can regulate the osteogenic differentiation of hBMSCs on the middle and late stages. However, vivo experiments are still needed to confirm that hsa_circ_0032600 in promoting osteogenic differentiation. And whether it can act as ceRNA, that is, the molecular sponge of miRNA, so as to reduce the effect of miRNA on gene expression still needs to be further investigated.

## CONCLUSIONS

In this article, we predict that the role of circ-LTBP2, which is shared by hsa-miR-24-3p, hsa-let-7b-5p and hsa-miR-5195-3p in osteogenic differentiation, still needs further experimental investigation. By using the prediction of the circRNAs that interact with 12 mRNAs and eight miRNAs related to osteogenesis, it is necessary to further explore whether therelevant circRNA-miRNA-mRNA interactions function as sponges during osteogenic differentiation.

In summary, our research firstly first involved biological information prediction. According to the KEGG analysis of circRNA parent genes, we identified 30 circRNAs and eight miRNAs that form six ceRNA networks. Moreover, the circRNAs located in key positions in the networks were identified as being involved in the potential ceRNA mechanism of the circRNA-miRNA-mRNA network, and their expression differences were verified by QRT-PCR analysis. To the best of our knowledge, the role of hsa_circ_0032600 that was verified in this study has not yet been reported. Therefore, this study demonstrates that SMAT titanium material can promote hBMSC osteogenic differentiation by affecting circRNAs, and this material can be used clinically.

## ACKNOWLEDGEMENTS

We thank Mingshan Li for guiding us about the bioinformatics.

### Funding

This study was supported by the National Natural Science Foundation of China (No. 81970980), Liaoning Province, Colleges and Universities Basic Research Project (No. LFWK201717), Liaoning Provincial Key Research Plan Guidance Project (No. 2018225078), Liaoning Provincial Natural Science Foundation Guidance Project (No. 2019-ZD-0749), Shenyang Major Scientific and Technological Innovation Research and Development Plan (No. 19-112-4-027), Shenyang Young and Middle-aged Technological Innovation Talent Plan (No. RC200060), Shenyang National Laboratory for Materials Science (No. 2015RP04), the Second Batch of Medical Education Scientific Research Projects of the 13th Five-Year Plan of China Medical University (No. YDJK2018017). The funders had no role in study design, data collection and analysis, decision to publish, or preparation of the manuscript.

### Grant Disclosures

The following grant information was disclosed by the authors:
National Natural Science Foundation of China: 81970980.
Liaoning Province, Colleges and Universities Basic Research Project: LFWK201717.
Liaoning Provincial Key Research Plan Guidance Project: 2018225078.
Liaoning Provincial Natural Science Foundation Guidance Project: 2019-ZD-0749.
Shenyang Major Scientific and Technological Innovation Research and Development Plan: 19-112-4-027.
Shenyang Young and Middle-aged Technological Innovation Talent Plan: No. RC200060.
Shenyang National Laboratory for Materials Science: No. 2015RP04.
Medical Education Scientific Research Projects: YDJK2018017.

### Competing Interests

The authors declare that they have no competing interests.

### Author Contributions

- Shanshan Zhu performed the experiments, analyzed the data, prepared figures and/or tables, authored or reviewed drafts of the paper, and approved the final draft.
- Yuhe Zhu performed the experiments, analyzed the data, prepared figures and/or tables, authored or reviewed drafts of the paper, and approved the final draft.
- Zhenbo Wang conceived and designed the experiments, authored or reviewed drafts of the paper, and approved the final draft.
- Chen Liang performed the experiments, prepared figures and/or tables, and approved the final draft.
- Nanjue Cao analyzed the data, prepared figures and/or tables, and approved the final draft.
- Ming Yan analyzed the data, prepared figures and/or tables, and approved the final draft.
- Fei Gao analyzed the data, prepared figures and/or tables, and approved the final draft.

- Jie Liu conceived and designed the experiments, authored or reviewed drafts of the paper, contributed the equipment and analysis tools, and approved the final draft.
- Wei Wang conceived and designed the experiments, authored or reviewed drafts of the paper, and approved the final draft.

## Data Availability

The raw measurements are available in the Supplemental Files.

## Supplemental Information

Supplemental information for this article can be found online at http://dx.doi.org/10.7717/peerj.9292#supplemental-information.

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
