# Peer review of "Bioinformatics analysis and identification of circular RNAs promoting the osteogenic differentiation of human bone marrow mesenchymal stem cells on titanium treated by surface mechanical attrition"

_PeerJ, doi:10.7717/peerj.9292_

## Round 0.1 · original submission · Major Revisions

Please address the significant issues raised by both expert reviewers. Each comment should be individually addressed and overall, the manuscript requires copy editing to improve flow.

·

Basic reporting

The manuscript entitled “Bioinformatics analysis and identification of circle RNAs promoting the osteogenic differentiation of human bone marrow mesenchymal stem cells on titanium treated by surface mechanical attrition” by Shanshan Z et al, is appropriate for this journal audience. The authors identify discrete circRNAs that contribute to the osteogenic differentiation of hBMSCs on SMAT -related titanium surface. Overall, the authors have done a great work on analysis of cRNAs and used various in vitro experiments to apply this knowledge to future clinical use.
However, there remain many concerns that could be addressed to ensure appropriate interpretation and reproducibility of these results.

Experimental design

Major concerns
1. Figures - I would like to request the pictures of Alizarin red stained samples. Also, please include the SEM image of surface treated titanium sheets.
Line 240 – for the alizarin red staining, if the significant difference was found on the 14th day, there is no explanation on this at discussion part.
2. Line 149 – It looks like from the data, Bradfords (BCA) assay wasn’t used to normalize protein quantity before initiating ALP assay. We cannot assure an equal amount of protein was present on all of 24 wells. Looking at Figure 6, SMAT cells already had more cells than annealed group, which means ALP activity can be influenced by this leading to misinterpretation of the data shown in Figure 7.
Line 237 – for the detection of ALP activity, if the significant difference was found on days 3 and 5, there is no explanation on this. Please discuss.
3. Line 164 – why was qPCR only performed at 21 days? Perhaps you can include more early timeframes to ensure a causal analyses.
4. Line 318 – why would you ‘predict’ something in the conclusion? And why do you think this still needs further experimental investigation? I didn’t find that in discussion.

Validity of the findings

Minor concerns
1. At the abstract, why is has-circ-00326000 ‘most obvious’? This is not a good word to use.
2. Line 27 – What do you mean by ‘attractive’? I would suggest to be more specific on the words you choose.
3. Line 28 – ‘small’ doesn’t seem to be a proper word to be used here. Please change to more coherent word. ‘Better delay absorption of alveolar bone’ can be changed in a better phrase.
4. Line 38 – why is ‘high diffusion rate and high chemical reactivity’ an advantage?
5. Line 46 – what do you mean by ‘do not show immunological rejection’? Do you mean not at all or low rejection rate?
6. Line 113 – Typo error – 2 104 Hz.
7. Line 145 – specify what was osteogenic induction medium.
8. Line 266 – ‘They have many advantages’. This sentence seems very short and doesn’t seem to be coherent with the flow of the paragraph.
9. Line 321 – Typo error – ‘trelevant’
10. Overall, I would suggest the authors use a native-English speaker or service to review their paper so it is more professional.

Additional comments

This is a paper with a lot of significance if potential errors are corrected.

Reviewer 2 ·

Basic reporting

This manuscript mainly wants to use biological information prediction to analyze the interactions between miRNAs, circRNAs and circRNA-miRNA-mRNA that interac with osteogenic mRNAs. In addition, the authors investigate the effects of surface mechanical attrition treatment (SMAT) on hBMSC osteogenic differentiation by affecting circRNA. The experimental objective is clear, while the English writing is poor, especially the Result Section. The literature references are sufficient and the sufficient background is provided. However, the results are not very clear and disordered. The number of Figures and Tables is too many to understand this manuscript.

Experimental design

The experimental design of this manuscript is relatively reasonable. The authors first used bioinformatics analysis and literature information to predict osteogenesis-related miRNAs, circRNAs and ceRNA networks of osteogenesis-related circRNAs by TargetScan, miRDB and miwalk3.0, and circRNA by circBase. Then, the authors used the SMAT titanium experimental material and the annealed titanium control material to detect the osteogenic differentiation ability of hBMSCs on the sample surfaces. However, the authors did not mention any information about the circRNAs that they detected. What’s the reason for choosing these circRNAs?

Validity of the findings

It is weak of evidence on the phenotype and functions of selected circRNAs. It also needs to add enough validation experiments about the effects of circRNAs on the osteogenic differentiation of hBMSCs by SAMT titanium.

Additional comments

It is a good idea to study the relationship between circRNA function and SMAT titanium material that promote hBMSCs osteogenic differentiation. In this study, the authors use a series of public dataset to support their research work by bioinformatics analysis, but it also needs more experimental evidence to prove the conclusions.

Major revisions:
1, the authors need to give the details of circRNAs that choose in the ceRNA network, and directedly osteogenesis-related miRNA, mRNAs.
2, the authors need to reorganized the orders of the figures. First is experimental results of SMAT titanium promoting osteogenic differentiation of hBMSCs, the second is bioinformatics prediction of osteoporosis-related circRNAs, and the third is the experimental validation of circRNAs function.
3, As presented, the English writing is not acceptable for the journal. The English writing of this manuscript must be improved carefully. Moreover, the introduction of Result section is too simple, not very logical and lack of summary. Therefore, this manuscript needs careful editing by someone with expertise in technical English editing paying attention to English grammar, spelling, and sentence structure.

---

## Round 0.2 · accepted · Accept

The reviewer's comments have been adequately addressed. Congratulations!

·

Basic reporting

No comment

Experimental design

On Figure 3, please pick one of the sample from the each group that shows most dramatic changes between alizarin red staining.

Validity of the findings

In line 27, change 'attractive' to 'esthetic'.

Additional comments

Thank you for all your corrections.